# IDEA-DAC: Integrity-Driven Editing for Accountable Decentralized Anonymous Credentials

## ABSTRACT

Decentralized Anonymous Credential (DAC) systems are increasingly relevant, especially when enhancing revocation mechanisms in the face of complex traceability challenges. This paper introduces **IDEA-DAC**, a paradigm shift from the conventional revoke-and-reissue methods, promoting direct and **I**ntegrity-**D**riven **E**diting (IDE) for **A**ccountable **DAC**s, which results in better integrity accountability, traceability, and system simplicity. We further incorporate an Edit-bound Conformity Check that ensures tailored integrity standards during credential amendments using R1CS-based ZK-SNARKs. Delving deeper, we propose a unique R1CS circuit design tailored for IDE. This design imposes strictly $O(N)$ rank-1 constraints for variable-length JSON documents of up to $N$ bytes in length, encompassing serialization, encryption, and edit-bound conformity checks. Additionally, our circuits only necessitate a one-time compilation, setup, and smart contract deployment for homogeneous JSON documents up to a specified size. While preserving core DAC features such as selective disclosure, anonymity, and predicate provability, IDEA-DAC achieves precise data modification checks that operate without revealing private content, ensuring only authorized edits are permitted. In summary, IDEA-DAC offers an enhanced methodology for large-scale JSON-formatted credential systems, setting a new standard in decentralized identity management efficiency and precision.

## CCS CONCEPTS

• **Security and privacy** → **Privacy-preserving protocols**; • **Applied computing** → **Version control**; • **Theory of computation** → *Circuit complexity*.

## KEYWORDS

Integrity-driven Editing (IDE), Decentralized Anonymous Credential (DAC), Edit-bound conformity check

**ACM Reference Format:**
Anonymous Author(s). 2018. IDEA-DAC: Integrity-Driven Editing for Accountable Decentralized Anonymous Credentials. In *Proceedings of Make sure to enter the correct conference title from your rights confirmation emai (Conference acronym 'XX)*. ACM, New York, NY, USA, 13 pages. https://doi.org/XXXXXXX.XXXXXXX

## 1 INTRODUCTION

Credentials serve as attestations, confirming an individual's identity or qualifications. Anonymous credentials emerged to fuse this verification process with user privacy preservation [6]. Decentralized anonymous credentials (DACs) [13] advance this concept, allowing users to verify specific attributes without revealing their complete identity, while operating independently of any central governing entity. Within a DAC framework, there are several pivotal algorithms, such as *request, issue, prove, verify,* and *revoke* [11].

The emergence of DACs and Decentralized Identifiers (DIDs) proffers a shift towards self-sovereign, privacy-centric authentication, standing in contrast to the predominant centralized models [4]. This paradigmatic transition towards Self-Sovereign Identity (SSI) through DAC is gaining momentum, especially with the advancements in blockchain technology [14, 20, 26]. Recognizing this evolution, the World Wide Web Consortium (W3C) is making strides towards standardizing DID documents [30] and the data model for verifiable credentials [31]. These standards seek to establish foundational metadata for identifiers and enhance the robustness of the verification mechanisms in digital transactions. Crucially, W3C endorses the expression of verifiable credentials in JSON-structured formats [31]. Such a credential can encapsulate facets ranging from issuer identity and subject attributes to specific conditions like validity periods. Leveraging embedded digital signatures, this JSON-oriented approach ensures data authenticity and resistance against tampering. To illustrate, W3C furnishes an exemplar[1] where an entire JSON document serves as a credential. This document comprises components like the issuer's details, issuance timestamp, and subject information, all enveloped using the JSON Web Signature to vouchsafe the data's integrity and authenticity.

Traditionally, updates to credentials necessitate a revoke-and-reissue approach to ensure system integrity and prevent unauthorized modifications [11, 23, 24]. This two-step approach not only introduces computational redundancy but also necessitates an additional revocation check during every credential verification, amplifying the overhead. Further compounding the inefficiency, these prevailing revoke-and-reissue paradigms frequently neglect the criticality of editing integrity. Without an integrity mechanism, systems cannot ensure that each data field consistently adheres to its established norms or criteria. Such oversight undermines system resilience and exacerbates risks, especially when data fields are adjusted to accommodate real-world scenarios.

Given the identified limitations in existing systems, it becomes beneficial to refine the credential update mechanism and enhance its integrity. To this end, we introduce **IDEA-DAC**, a novel methodology that facilitates **I**ntegrity-**D**riven **E**diting (IDE) designed for **A**ccountable **D**ecentralized **A**nonymous **C**redentials. Contrary to traditional systems, IDEA-DAC enables edit directly to a JSON credential document, ensuring that security and trustworthiness

---

[1]https://www.w3.org/TR/vc-data-model/

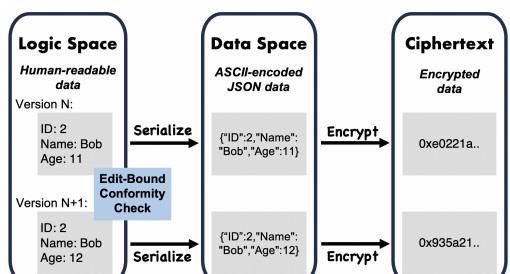

**Figure 1: Edit-bound Conformity Check Framework**

remain intact. Central to our approach is the application of zero-knowledge proofs that rigorously assess each edit against robust integrity standards, thereby presenting an optimized and holistic framework for the management of JSON credentials.

Furthermore, we introduced an edit-bound conformity check to ensure the integrity of credentials during modifications, utilizing ZK-SNARKs built upon Rank-1-Constraint-System (R1CS). The edit-bound conformity check introduces a framework that emphasizes integrity checks in the logic space rather than directly on the data space, which allows for fine-grained control over human-readable data, as illustrated in Figure 1. Subsequently, data from this logic space undergoes serialization into the data space, followed by encryption into ciphertext. This encrypted form is then suitable for publication. We formulated a distinctive R1CS circuit designed for IDE. This design mandates $O(N)$ rank-1 constraints for JSON documents with variable lengths, up to a maximum of $N$ bytes, covering serialization, encryption, and edit-bound conformity checks. Crucially, our circuits requires only a singular compilation, setup, and smart contract deployment, catering to homogeneous JSON documents up to a predefined size.

In our experimental assessment, we implemented IDEA-DAC and evaluated its performance using a real-world scenario: academic profiles of Ph.D. students, which demonstrates the potential data types and associated integrity rules in a JSON credential. We primarily gauged our algorithm's efficiency using three metrics: number of rank-1 constraints, proving time, and verification time. Our results suggest a linear correlation between the credential file size and number of constraints. Additionally, the proving time stays pragmatically efficient, and the verification time remains constant.

While preserving DAC security and privacy properties, such as selective disclosure, anonymity, and predicate provability, IDEA-DAC ensures precise data modification checks. These checks operate without exposing sensitive data during credential edits, thereby strengthening the system against any unauthorized changes. The main contributions of IDEA-DAC can be delineated as:

- **Integrity-Driven Editing (IDE):** IDEA-DAC introduces a new mechanism to a DAC system named Integrity-Driven Editing (IDE). It strengthens editing integrity and traceability, harnessing the power of ZK-SNARKs built upon R1CS circuits. Moreover, IDEA-DAC introduces edit-bound conformity checks as specialized sub-circuits. These checks impose concrete integrity standards during credential amendments. As opposed to generic methods, our approach ensures that all credential adjustments strictly comply with established rules. Via our circuit design, these integrity checks seamlessly operate within the logic structure of the data, all the while upholding the privacy of credentials.

- **Advanced JSON serialization Circuit:** IDEA-DAC introduces a pioneering R1CS circuit design specifically tailored for JSON serialization, achieving $O(N)$ rank-1 constraint complexity for JSON documents up to $N$ bytes in size. This innovative design not only lays the groundwork for deeper investigations into R1CS circuit architectures and JSON-compatible systems but also ensures a one-time setup with inherent scalability. By adeptly avoiding potential exponential complexities, it proves particularly robust for handling expansive JSON datasets, such as repositories detailing an academic's comprehensive publication history.

## 2 RELATED WORK

Decentralized identity research has had a multitude of projects from both academia and industry. Many Decentralized Identity (DID) initiatives commonly feature Predicate Provability, enabling users to demonstrate compliance without exposing personal data. The W3C DID standard acts as a key reference; systems are typically classified as compliant or non-compliant. Compliant systems such as Candid [21], SpruceID [28], and Verite [8] exist, but only Candid prioritizes privacy-preserving, granting data access solely upon user approval. Contrastingly, non-W3C compliant systems like zk-creds [24] and Coconut [27] also emphasize user privacy. However, a consistent challenge for these systems is the accurate tracking of Precise Data Modification Checks. Distinctly, IDEA-DAC, showcased in Table 1, amalgamates features such as Privacy-preserving, Predicate Probability, W3C Conformity, and precise data change monitoring. Notably, while Coconut deploys a decentralized strategy against malicious issuers, IDEA-DAC utilizes edit-bound conformity checks, optimizing the oversight of issuer activities.

Another branch of research that aligns with our work delves into secure data modifications and privacy-preserving verification, especially when multiple parties are involved. For instance, in the area of image editing, Naveh et al. presented PhotoProof [22]. Their solution allows certain image alterations while ensuring the edited image remains traceable and authentic. Meanwhile, zkDocs [1] emphasizes the secure verification of documents in contexts like mortgage applications. While these approaches have demonstrated feasibility on real-world data, they pose implementation challenges. The underlying ZK circuits demand frequent adjustments: any change in the underlying data requires recompilation, setup, and redeployment of the circuit. To illustrate, even a minor image adjustment, such as cropping from $512 \times 512$ to $384 \times 384$, would require a brand new circuit specific to the modified size. Additionally, while PhotoProof can track pixel-wise transformations, zkDocs lacks the ability to examine fine-grained edits within the circuit, largely due to its representation of data that isn't interpretable by humans. Our proposed IDEA-DAC addresses these challenges. It features a circuit design that requires setup only once, suitable for variable-length JSON documents, and is adept at ensuring detailed edit integrity checks, leveraging an advanced R1CS circuit framework.

## 3 PRELIMINRIES

In this section, we delve into the foundational concepts necessary for constructing the IDEA-DAC system. Specifically, we introduce ZK-SNARKs, detail the structure and significance of R1CS, and explore the Hint strategy and its implications.

| | | Candid [21] | SpruceID[28] | Verite[8] | zk-creds [24] | Coconut [27] | **IDEA-DAC** | PhotoProof [22] | ZKDocs [1] |
|---|---|---|---|---|---|---|---|---|---|
| DAC Security & Privacy Properties | Privacy-preserving | ✔ | ✗ | ✗ | ✔ | ✔ | ✔ | | |
| | Predicate Provability | ✔ | ✔ | ✔ | ✔ | ✔ | ✔ | | |
| | W3C Conformity | ✔ | ✔ | ✔ | ✗ | ✗ | ✔ | | |
| | Precise Data Editing Checks | ✗ | ✗ | ✗ | ✗ | ✗ | ✔ | | |
| Multi-party Privacy-preserving Information Sharing & Verification | Verifiable Information Editing | | | | | | ✔ | ✔ | ✔ |
| | Fine-grained Editing Check | | | | | | ✔ | ✔ | ✗ |
| | One-time Setup ZK Circuit | | | | | | ✔ | ✗ | ✗ |
| | Programmable Editing Bound | | | | | | ✔ | ✔ | ✗ |

**Table 1: Comparison between IDEA-DAC and other related systems in terms of DAC Security & Privacy Properties and Multi-party Privacy-preserving Information Sharing & Verification**

## 3.1 ZK-SNARKs

Zero-Knowledge Proofs (ZKPs) [15, 16] are a cryptographic method by which one party (the prover) can prove to another party (the verifier) that they know a value $x$, without conveying any information apart from the fact that they know the value $x$. A formal definition of ZKP can be found in Appendix B.

Zero-Knowledge Succinct Non-Interactive Argument of Knowledge (ZK-SNARKs) [5, 12, 17, 33, 35] are a special form of ZKPs that have the properties of being non-interactive and succinct. Non-interactivity means that the proof consists of a single message from the prover to the verifier. Succinctness means that the size of the proof is small (polylogarithmic in the size of the statement being proven), and verification of the proof is fast (also polylogarithmic).

## 3.2 Rank-1 Constraint System (R1CS)

A Rank-1 Constraint System (R1CS) over a field $\mathbb{F}$ serves as a foundational arithmetic representation in the construction of ZKP protocols. Its structure provides a systematic way to express and verify computational statements without revealing any information about the inputs, other than the fact that they satisfy the given statement. Formally, an R1CS over a field $\mathbb{F}$ is described as a triple $(A, B, C)$, where each of $A$, $B$, and $C$ are $n \times m$ matrices. Here, $n$ denotes the number of constraints, while $m$ represents the number of wires. An assignment $\mathbf{x} \in \mathbb{F}^m$ is said to satisfy the R1CS if and only if the following equation is true for all $i$: $\langle \mathbf{a}_i, \mathbf{x} \rangle \cdot \langle \mathbf{b}_i, \mathbf{x} \rangle = \langle \mathbf{c}_i, \mathbf{x} \rangle$, where $\mathbf{a}_i$, $\mathbf{b}_i$, and $\mathbf{c}_i$ represent the $i$-th rows of matrices $A$, $B$, and $C$ respectively, and $\langle \cdot, \cdot \rangle$ denotes the dot product. In the realm of ZKP protocols, R1CS plays a pivotal role in constructions such as ZK-SNARKs (e.g., Groth16 [17], Marlin [7], Aurora [3], Orion [34]) and recent folding schemes (e.g., Nova [19], SuperNova [18]). These protocols exploit the succinctness and efficiency of R1CS to provide proofs that are both compact and quick to verify, thereby enabling a myriad of applications in privacy-preserving transactions, secure computations, and blockchain scalability solutions.

## 3.3 Hint

R1CS, inherently designed to verify constraints for inputs, extends its capabilities beyond mere value computations within a circuit. It introduces two primary methods for calculating values designated for other circuit sections. The first is a direct approach that utilizes circuit arithmetic for computation. However, when this direct method becomes inefficient or cumbersome, an alternative strategy, termed **Hint**, is preferred. The "Hint" method allows the value to be computed off-circuit, and validated in-circuit. This implies that the prover can input the value directly into the circuit, with constraints in place to ensure its accurate computation. To illustrate, consider the computation of $y = f(x)$. If constructing $f(\cdot)$ within the circuit is challenging, an alternative is to use a function $g(x, y)$

that verifies the correctness of this computation in a more efficient manner. For instance, to compute the inverse of $x \in \mathbb{Z}_p$, one might directly compute $y = x^{p-2}$ in-circuit with $\lceil \log p \rceil$ multiplications. However, with the "Hint" approach, by directly inputting $y$, only a single rank-1 constraint is needed to ensure $x \cdot y = 1$. Effectively deploying the "Hint" strategy in cases where direct computation is onerous can result in significant in-circuit computational savings, leading to enhanced proving and verification times.

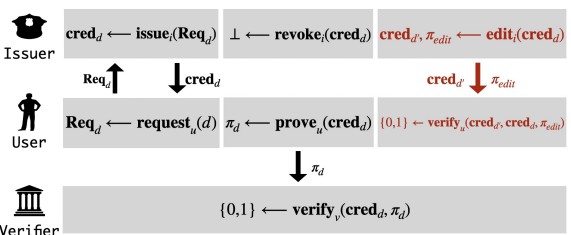

**Figure 2: IDEA-DAC Functionalities**

## 4 IDEA-DAC

Decentralized anonymous credentials (DACs) [13] are essential for decentralized identity systems. DACs allow users to get verified without showing their actual identities, balancing privacy with trust. In simple terms, DACs let users hold multiple credentials, each approved by the issuer's private key, eliminating intermediaries during the verification process.

In a DAC system, three main players exist: issuers, users, and verifiers. Users ask for credentials by sharing their data and any other needed information. Issuers, using their private key, grant these credentials and remain the rights to revoke them. Users can then show these credentials to verifiers, who can check their authenticity without contacting the issuer directly.

However, existing DAC systems overlook the integrity of editing. Within the current DAC's method of revoking and reissuing as a form of editing, maintaining the integrity of edits is challenging. Furthermore, editing goes beyond mere content alterations; it necessitates setting distinct integrity rules for different editors in different use cases. In response to this challenge, we present IDEA-DAC: a system that guarantees editing integrity in JSON-formatted DACs through edit-bound conformity checks, as depicted in Figure 2. This section delves into the formal definitions of JSON credentials and the associated edit-bound conformity checks.

## 4.1 JSON Credentials

A credential $P$ is represented as a JSON document, comprised of various fundamental items. These items include strings, numbers, arrays, and dictionaries. For the purpose of subsequent definitions, we introduce a universal object set, $O$. It is formally defined as

$O := S \cup N \cup A \cup D$, where $S, N, A,$ and $D$ stand for the sets of all strings, numbers, arrays, and dictionaries, respectively. This definition of $O$ will be utilized in the ensuing sections.

**Definition 4.1 (String).** A *String* item is denoted by $r \in \Sigma^*$, where $\Sigma$ is the set of all the possible characters.

To encompass most practical scenarios, we assume all characters to be represented as ASCII-encoded byte values; thus, $\Sigma = [0, 255]$.

**Definition 4.2 (Number).** Define a *Number* item $r$ as $r \in \{0, 1, 2, 3, 4, 5, 6, 7, 8, 9\}^*$.

The definition for numbers can be generalized to encompass positive integers. Furthermore, it can be extended to accommodate various numerical representations, including negative integers (denoted by $-r$), floating-point numbers (denoted by $(-)r_1.r_2$), and scientific notations (denoted by $(-)r_1.r_2e(-)r_3$).

**Definition 4.3 (Array).** An array is $a \in O^*$, which means $a$ is an ordered tuple whose elements are all elements in $O$.

In an array, an element can be of types such as a string, integer, dictionary, or even another array. The length of the array can be adjusted by adding or removing items within it.

**Definition 4.4 (Dictionary).** A dictionary $d$ is a set of key-value pairs with the following properties:

$$d \in D \iff \exists n \in \mathbb{Z}^+, d = \{(k_1, v_1), (k_2, v_2), \ldots, (k_n, v_n)\},$$
$$k_i \in S, \quad v_i \in O \quad \text{for each } i \in [n],$$

A dictionary within a credential comprises multiple key-value pairs; the key must always be a string, whereas the value can be a string, integer, array, or even another dictionary. Conventionally, a JSON credential $P$ is represented as a dictionary, i.e., $P \in D$.

## 4.2 Edit-bound Conformity Check

Every legitimate editing activity must adhere to several *integrity rules*, each represented as a function $\sigma$. These rules ensure that the editor follows the appropriate rules to undertake such edits. The integrity verification comes in two variants: the target-only check and the source-target differential check.

**Definition 4.5 (Target-only check).** A target-only check is a function $\sigma_\theta(r) \rightarrow \{0, 1\}$, where $r$ is the edited credential item and $\theta$ is some pre-defined parameters.

The combination of $\sigma$ and $\theta$ defines an *integrity rule* specific to an editor. As an example, an editor might be constrained to an *integrity rule* for a *String* credential item, termed as "One of the Set". This rule restricts the editor to only modify the item to a new string $r'$ such that $r' \in \theta$. Here, $\theta$ denotes a pre-defined set of strings allowable for this particular editor. Thus, the pairing of $\sigma$ with $\theta$ characterizes a distinct *integrity rule* for that editor.

At times, merely examining the target credential is inadequate for regulating an editing activity. A *source-target differential check* assesses the disparity between the pre- and post-edited credentials.

**Definition 4.6 (Source-target differential check).** A source-target differential check is a function $\sigma'_\theta(r, r') \rightarrow \{0, 1\}$, where $r$ and $r'$ stand for pre- and post-edited credential items.

Just as with target-only checks, an *integrity rule* is defined by the combination of $\sigma'$ and $\theta$. For instance, an editor might be subject to an integrity rule for an array item, termed "Append Only". This rule dictates that the editor is only permitted to append elements to the array item $r$. If the augmented array is denoted as $r'$, $\sigma'_\theta(r, r') = True \iff r_i = r'_i, \forall 0 \leq i < |r|$, where $|r|$ is the length of the array item $r$. In such a situation, $\theta$ is always $\perp$, meaning that this rule does not need additional parameters.

These two checks solely verify if the editing activity adheres to the appropriate integrity rules. However, if an editor intends to publish the edited credential item as a public record, additional checks must be passed. We denote the checks that need to be conducted prior to publishing content as the *encoding and encryption check $\omega$*, which assesses the correctness of the serialization and encryption procedures. It will grant approval if and only if both the serialization and encryption processes are executed correctly.

**Definition 4.7 (Encoding and Encryption check).** An encoding and encryption check is a function: $\omega(r, c, k) \rightarrow \{0, 1\}$ that verifies if $c = Enc_k(Serialize(r))$, where $r$ is the credential and $Enc, Serialize$ is the symmetric encryption algorithm and the serialization algorithm respectively.

Finally, we define the *edit-bound conformity check*, which encompasses all the checks above.

**Definition 4.8 (Edit-bound Conformity Check).** Denote $\lambda$ as the security parameter. The *edit-bound conformity check* is represented by the function $\phi(r, r', c, c', k, \Sigma, \Sigma') \rightarrow \{0, 1\}$, where $r$ and $r'$ denote the pre- and post-edited items, respectively, and $c$ and $c'$ are their corresponding ciphertexts. Additionally, $k$ is the encryption key, known to both the user and editors, and $\Sigma, \Sigma'$ stands for set of all the applied target-only checks and the source-target differential checks. Note that both $c$ and $c'$ are encrypted using the same key $k$. The function $\phi$ possesses the subsequent properties:

- **Completeness:** If $(\forall \sigma_\theta, \sigma'_\theta \in \Sigma, \Sigma', \sigma_\theta(r') = 1, \sigma'_\theta(r, r') = 1) \wedge (\omega(r, c, k) = \omega(r', c', k) = 1) : \mathbf{Pr}\{\phi(r, r', c, c', k, \Sigma, \Sigma') = 1\} = 1$
- **Soundness:** If $(\exists \sigma_\theta \in \Sigma, \sigma_\theta(r') = 0) \vee (\exists \sigma'_\theta \in \Sigma', \sigma'_\theta(r, r') = 0) \vee (\omega(r, c, k) = 0) \vee (\omega(r', c', k) = 0) : \mathbf{Pr}\{\phi(r, r', c, c', k, \Sigma, \Sigma'\} = 1) \leq negl(\lambda)$
- **Privacy-preserving:** for any Probabilistic Polynomial Time (PPT) distinguisher $\mathcal{D}$, the probability that $\mathcal{D}$ can correctly distinguish which of the two credentials has been randomly sampled, without knowledge of the key $k$, is bounded as: $\mathbf{Pr}\{\mathcal{D}(r, \hat{r}) = 1 \mid c, \hat{r} \xleftarrow{\$} R\} = \frac{1}{2} + negl(\lambda)$, where $R$ is the set of all possible credentials. This property applies to both the pre- and post-edited credentials.

## 5 INTEGRITY-DRIVEN EDITING (IDE)

In this section, we delve into the intricate R1CS circuit design behind Integrity-Driven Editing (IDE) for IDEA-DAC. With a focus on modularity and efficiency, the IDE circuit is crafted to validate every edit made to a JSON document against strict integrity rules, all while requiring setup only once. Through advanced serialization, encryption, and efficient batch merging, it guarantees secure and optimal in-circuit document processing. Coupled with stringent conformity checks, this design assures both the accuracy and integrity of document edits, making it a cornerstone for the secure handling of large-scale JSON credentials in real-world scenarios.

## 5.1 R1CS Primitives

The Rank-1 Constraint System (R1CS) is essential for converting certain logic structures. To effectively harness the power of R1CS, we introduce several foundational primitives. A critical element in this process is the **linear combination**, which underpins the necessary composability for circuit logic.

**Definition 5.1 (Linear combination).** A linear combination, denoted as $l = [(c_i, x_i) \mid i = 1, 2, \ldots, n]$, comprises an array of 2-tuples $(c_i, x_i)$ with arbitrary length $n \in \mathbb{Z}^+$. Here, $c_i \in \mathbb{F}$ represents the coefficient, while $x_i \in \mathbb{Z}_{\geq 0}$ is the wire ID in R1CS, denoting the $x_i$-th element in the assignment vector **x**.

Notably, the variable ID 0 signifies the constant 1. Following this definition, the conversion of a linear combination into rows within the matrices $A$, $B$, and $C$ is intuitive. To streamline our discussions, we'll use **linear combination** and **variable** interchangeably henceforth. Based on the foundation of linear combinations, we present the subsequent R1CS primitives:

- $a_{[N]} \leftarrow$ **NewArray**($N$): Initializes an array of size $N$.
- $c \leftarrow$ **Add**($a, b$): Combines the terms of $a$ and $b$ to produce a new linear combination. No new wires or constraints are added.
- $c \leftarrow$ **Sub**($a, b$): Produces a linear combination by subtracting $b$ from $a$. To achieve this, all coefficients in $b$ are negated in $\mathbb{F}$. No additional wires or constraints are introduced.
- $c \leftarrow$ **Mul**($a, b$): Represents the product of two variables. If either variable is constant, the coefficients of the other are scaled. Otherwise, it creates $c = [(1, \text{newID})]$ and a constraint $a \cdot b = c$.
- $c \leftarrow$ **Or**($a, b$): Computes the logical OR between two Boolean-constrained variables. If both are Boolean-constrained, it results in the constraint $a \cdot b = a + b - c$.
- $c \leftarrow$ **And**($a, b$): Computes the logical AND for two Boolean variables, which is synonymous with **Mul**($a, b$).
- $b \leftarrow$ **IsZero**($a$): Determines if variable $a$ is zero. Instead of using $\lceil \log |\mathbb{F}| \rceil$ constraints for direct computation, it uses three constraints based on hint. Specifically, the constraints are: $b \cdot (1-b) = 0$, $a \cdot b = 0$, and, given the inverse $c$ of $a + b$, $(a + b) \cdot c = 1$.
- $b \leftarrow$ **IsNotZero**($a$): Checks if variable $a$ is non-zero. This is derived from **IsZero**($a$) using the operation $b \leftarrow$ **Sub**(1, **IsZero**($a$)).
- $c \leftarrow$ **IsEqual**($a, b$): Checks if $a$ and $b$ are equal, implemented using **IsZero**(**Sub**($a, b$)) with three constraints.
- $c \leftarrow$ **Select**($s, r_0, r_1$): Outputs $r_0$ if $s = 1$ and $r_1$ otherwise. It introduces $c = [(1, \text{newID})]$ and the constraint $s \cdot (r_0 - r_1) = c - r_1$.
- **Assert**($a == b$): Ensures $a$ and $b$ are equal in the constraint system using the constraint $a \cdot 1 = b$.
- $b \leftarrow$ **WithinBinary**($a, N$): Verifies if variable $a$ has a bit-size of at most $N$. It inputs the bit decomposition $bit_{[N]}$ of $a$ and uses $N + 3$ constraints, specifically: $bit[i] \cdot (1 - bit[i]) = 0, \forall i \in [N]$, and checks the decomposition with **IsEqual**(($\sum_{i=0}^{n-1} 2^i bit[i]$), $a$).
- $r \leftarrow$ **RO**(...): Utilizes an in-circuit random oracle to generate a pseudo-random variable derived from existing circuit variables. This is enforced using collision-resistant ZK-friendly hash functions like MiMC, resulting in efficient R1CS construction.

## 5.2 Length-prepadded String (LPS)

To standardize a one-time setup for homogeneous documents, the circuit must accommodate variable-length inputs up to a maximal threshold. This flexibility allows for efficient version control when documents of the same kind undergo changes, such as increased string lengths or added array items. By introducing redundancy into the circuit structure, we can mitigate the need for frequent re-compilations, setups, and deployments whenever data undergoes modifications. Thus, we present a length-prepadded representation for strings. This encoding embeds the actual string length directly into the circuit, facilitating efficient string processing. Given that ASCII characters are byte-encodable, each byte is treated as an individual variable for easier string operations.

**Definition 5.2 (Length-prepadded String (LPS)).** An LPS, $s$, with a maximum byte-length $N$, is an array of size $N + 1$ described as $s = [l, s_0, s_1, \ldots, s_{l-1}, D, \ldots, D]$. Here, $l, s_i \in \mathbb{F}$ with $s_i \in [0, 255]$ and $l < N$. The symbol $D$ signifies a dummy constant excluded from valid string characters, such that $D \in \mathbb{F}$ and $D \notin [0, 255]$. For notation simplicity, we use $l \| s_{[N]}$ to represent the string.

To further streamline string operations, we introduce two primitives for handling dummy values:

- $b \leftarrow$ **IsDummy**($a$): Determines if variable $a$ represents a dummy character in a string, achieved using **IsEqual**($a, D$).
- $b \leftarrow$ **IsNotDummy(IND)**($a$): Verifies if variable $a$ is not a dummy character, computed via **Sub**(1, **IsDummy**($a$)).

## 5.3 LPS Operations

*5.3.1 Range Check.* To ensure that string inputs to the circuit are valid, every string variable must either be a byte or the dummy value $D$. Specifically, $s[i] < 256$ or $s[i] = D$ for all $i = 0, 1, \ldots, N-1$. For this purpose, we propose an $O(N)$-sized range check circuit: **Assert**$\left( \sum_{i=0}^{N-1} (\textbf{WithinBinary}(s[i], 8) + \textbf{IsDummy}(s[i])) == N \right)$, which necessitates $14N + 1$ constraints. The correctness of this construction is evident when observing that both **WithinBinary** and **IsDummy** cannot simultaneously be true. If any $s[i]$ falsifies both, the sum will be less than $N$. Therefore, the assertion mandates that at least one of these values be true for each $s[i]$.

*5.3.2 Legitimacy Check.* For operations to be correctly executed later, strings input to the circuit must adhere to the LPS format. To verify this for all strings up to a maximum length $N$, we propose an $O(N)$ legitimacy check. Given an LPS $l \| s_{[N]}$, the legitimacy check validates three aspects: 1) $s[i] \neq D$ for $0 \leq i < l$; 2) $s[i] = D$ for $l \leq i < N$; 3) $l < N$. While this may seem straightforward, accounting for variable $l$ that might assume any value in $\mathbb{F}$ complicates matters. A naive approach involves $O(N)$ in-circuit variable comparisons, resulting in $O(N \log N)$ constraints, which is sub-optimal. We present an optimized legitimacy check that needs only $O(N)$ constraints. The circuit computes a tag $c[i]$ for each variable: $c[i] = l - i - 1$ if $s[i] \neq D$, and $i + 1 - l$ otherwise. This requires $4N$ constraints. Then, we establish the following constraints based on $c[i]$, also taking into account if the end of the actual string has been reached: $c[i] == c[i-1] - 1$ if the end hasn't been reached, and $c[i] == c[i-1] + 1$ otherwise. This implementation requires $7N - 6$ constraints. The string ends when the first index $i$ has $c[i]$ as a dummy variable. This is tracked using a Boolean variable, requiring $N$ constraints. Then, considering: $0 < l < N \iff \exists 0 \leq i < N, c[i] = 0$. We need to ensure either: 1)

only one 0 exists in $c[i]$; or 2) $l = 0$. This demands $3N+4$ constraints. An optimized legitimacy check algorithm is shown in Algorithm 2.

### 5.3.3 Merge Check.

During JSON credential serialization, two LPS strings must be concatenated while retaining format integrity. Constructing a generic circuit capable of merging strings of various lengths poses challenges. It's essential to design a multipurpose circuit that merges any two strings efficiently and correctly.

The naive approach requires $O((N_a+N_b)^2)$ constraints to merge two LPS strings of maximal lengths $N_a$ and $N_b$, as shown in Algorithm 6. We propose a hint-based optimization that needs only $O(N_a + N_b)$ constraints. Instead of using a multiplexer, the prover inputs the merged LPS directly. The circuit then verifies its correctness via various constraints. Given two LPS structures, $s_a =$ aLen$||$a$_{[N_a]}$ and $s_b =$ bLen$||$b$_{[N_b]}$, the prover inputs the result LPS $s_c =$ cLen$||$c$_{[N_a+N_b]}$ directly into the circuit. We validate $s_c$ by ensuring it's correctly formatted and is the exact merged LPS of $s_a$ and $s_b$ through the following checks:

(1) Assert that the length of the merged string is the sum of the lengths of both strings: **Assert**(cLen == **Add**(aLen, bLen)).
(2) Ensure the range check of the merged string: **RangeCheck**( c$_{[N_a+N_b]}$), referencing the same circuit as in Section 5.3.1.
(3) Invoke an in-circuit random oracle to produce a pseudo-random value $r$ and validate the equation:

$$\prod_{i=0}^{N_a-1} [r - (256(i+1) + a[i])\mathbf{IND}(a[i])]$$
$$\times \prod_{i=0}^{N_b-1} [r - (256(i+1+\text{aLen}) + b[i])\mathbf{IND}(b[i])] \quad (1)$$
$$= \prod_{i=0}^{N_a+N_b-1} [r - (256(i+1) + c[i])\mathbf{IND}(c[i])],$$

Define sets $S_1 = \{(256(i+1) + a[i])\mathbf{IND}(a[i])\} \cup \{(256(i+1+\text{aLen})+b[i])\mathbf{IND}(b[i])\}$ and $S_2 = \{(256(i+1)+c[i])\mathbf{IND}(c[i])\}$. Each variable in a$_{[N_a]}$ and b$_{[N_b]}$ is coupled with its target index in the merged string. The lowest 8 bits store the byte value, while the subsequent bits indicate the target index starting from 1. This ensures the uniqueness and non-zero nature of each non-dummy element in sets $S_1$ and $S_2$. Due to the range check constraints and unique index bits of the variables, we ensure that the first aLen + bLen variables in $S_2$ match those in $S_1$. Given $s_a$ and $s_b$ are both legitimate LPS as checked initially, and there are exactly aLen+bLen non-zero terms in $S_1$, we infer that all the other variables in $S_2$ are dummy variables, ensuring the legitimacy of $s_c$. Finally, according to Schwartz–Zippel lemma [25], this check passes with a negligible soundness error of $\frac{N_a+N_b}{|\mathbb{F}|}$. This final check utilizes $|\mathbf{RO}| + 10(N_a + N_b) - 2$ constraints, a great improvement over the naive method when employing ZK-friendly hash functions as the random oracle.

## 5.4 Serialization

Drawing from the foundational LPS operations, we design a JSON serialization circuit, aligning with the definition in Section 4.1. By predetermining maximal length parameters for Number, String, and Array, we eliminate repetitive re-compilations, setups, and deployments. While changing value types within a Dictionary poses challenges, we address this by presuming fixed value types given the typical static nature of JSON document structures. This strategy allows flexibility with arbitrary keys and values, ensuring a one-time setup for homogeneous documents. We subsequently detail the serialization circuit **Encode** for each JSON data type.

### 5.4.1 Encoding Number.

The process of encoding a number mandates its decimal decomposition within the circuit. This operation, when approached traditionally, can be computationally costly. To ameliorate this, we employ a strategy similar to the one in **WithinBinary** for bit-decomposition using hints. This strategy involves computing the decomposition outside the circuit and subsequently verifying its accuracy within the circuit. For a given number $x$ with a maximum of $N$ digits, the prover inputs a length-prepadded decimal decomposition, represented as len$||$dec$_{[N]}$ in big-endian format, with trailing 0s serving as dummy digits. The decomposition's accuracy is confirmed using three dedicated sub-circuits:

(1) **RangeCheckDecimal**(dec$_{[N]}$): This validates that each dec$[i]$ lies in $[0, 9]$ for all $i \in [N]$: **Assert**$\big(\sum_{i=0}^{N-1}(\mathbf{IsEqual}(s[i], 8) + \mathbf{IsEqual}(s[i], 9) + \mathbf{WithinBinary}(s[i], 3)) == N\big)$, necessitating $12N + 1$ constraints. The correctness of this method resembles that for the LPS range check in Section 5.3.1.
(2) **Decomposition Check**: Utilizing a remLen variable to track the residual length, and an isEnd variable to signify the termination of the current digit sequence, we reconstruct the digits in big-endian format to match the original number using: 1) sum = **Select**(isEnd$[i]$, sum, **Mul**(sum, 10)); 2) sum = **Add**(sum, dec$[i]$). Then, we use an assertion, **Assert**(sum = $x$), to validate the correctness of the decomposition. It's noteworthy that this mechanism inherently ensures $x < 10^N$, given the nonexistence of an $N$-digit decomposition for $x$ when $x \geq 10^N$.
(3) **Legitimacy Check**: This ascertains that the decomposition is in valid format, ensuring all trailing digits are zeroes, represented by: **Assert**(**And**(isEnd$[i]$, **IsNotZero**(dec$[i]$)) == 0).

The final step involves translating the digits into an ASCII-encoded LPS as: out$[i]$ = **Select**(isEnd$[i]$, $D$, **Add**(dec$[i]$, 48)). Collectively, these checks and conversions demand $O(N)$ rank-1 constraints. A complete algorithm is provided in Algorithm 1.

### 5.4.2 Encoding String.

Before we initiate the encoding of strings, it's essential to validate that the strings adhere to the expected LPS format. This validation is achieved through the range and legitimacy checks as described in Sections 5.3.1 and 5.3.2. After ensuring they are in the correct LPS format, we then merge a double quotation mark at both the beginning and end of the LPS.

### 5.4.3 Encoding Array.

When encoding an array that contains various objects, it's crucial to exclude any empty elements from the serialized LPS. To this end, we implement the **IsEmpty** function to ascertain the status of each object within the array. For a Number $x$, we designate a value $D'$, such that $D' > 10^{\text{maxDigit}}$, to represent its emptiness, with a slight modification of the number encoding circuit. For an LPS structure, $l||s_{[n]}$, the emptiness is determined by the condition $l = 0$. Both Arrays and Dictionaries are assessed recursively to determine their emptiness. Given these mechanisms, the encoding process for an Array $a_{[n]}$ can be outlined as: 1) tmpLPS = **Merge**(oldLPS, **Encode**(a[i])); 2) newLPS = **SelectArray**(**IsEmpty**(a[i]), oldLPS, tmpLPS), where **SelectArray** is **Select** over each individual variable.

*5.4.4 Encoding Dictionary.* Encoding a Dictionary closely mirrors the Array encoding process, but there's a key distinction to note: Dictionary values are typified from the beginning. Therefore, when invoking the **Encode** function, both the key and its paired value are encoded. This approach offers flexibility, allowing documents to omit certain predefined Dictionary fields, while still adhering to a one-time setup. To sum up, given every JSON document inherently represents a Dictionary, the serialization process begins with this primary Dictionary, and then recursively proceeds to ensure accurate serialization of the entire JSON document.

## 5.5 Encryption

Upon obtaining the serialized JSON string $l||s_N$ as an LPS, we proceed to encrypt it within the circuit using an encryption key. This encryption conceals the internal data before it is published. Although ZK-friendly encryption algorithms like MiMC can adeptly encrypt field elements using minimal rank-1 constraints, encrypting the LPS byte-by-byte could result in a vast ciphertext. This arises since every byte is encrypted into a separate field element, potentially consuming up to 254 bits in the BN254 elliptic curve which is supported by native EVM pre-compiles. To address this, we emphasize compressing the LPS bytes before encryption. Each byte occupies a maximum of 8 bits, making it efficient to merge every $\lfloor \frac{254}{8} \rfloor = 31$ bytes into a single variable, achieving lossless compression. Additionally, it's crucial to encrypt only the actual length bytes, which necessitates additional constraints to manage the dummy values. The **Compress** procedure constructs compressed variables $m_{[M]}$, where $M = \lceil \frac{N}{31} \rceil$, and a dummy indicator isDummy$_{[M]}$ as: 1) $m[i] = \sum_{j=31i}^{\min(N,31(i+1))} 2^{8(j-31i)} s[j] \cdot \textbf{IND}(s[j])$; 2) isDummy$[i] = $ **IsZero**$\left( \sum_{j=31i}^{\min(N,31(i+1))} \textbf{IND}(s[j]) \right)$. Then, the **Encrypt** operation generates the ciphertext $c_{[M]}$ using the secret key $k \in \mathbb{F}$: $c[i] = $ **Select**(isDummy$[i]$, 0, **MiMC**$(k, m[i])$). We assign 0 to dummy variables to simplify the array comparison process in Section 5.8.

## 5.6 Edit-bound Conformity Check Circuit

Once all the data in the JSON document is logically represented in the circuit inputs, incorporating checks on this data becomes relatively straightforward, requiring only a modest number of rank-1 constraints. For instance, to ascertain whether a Number lies within a specific range, merely two in-circuit variable comparisons suffice. Verifying the correctness of a string's format can be accomplished via character-by-character comparison or a range check.

Nonetheless, each edit-bound check circuit is intrinsically tailored to its specific use cases. Attempting to design a universal circuit capable of managing every conceivable check isn't pragmatic. To retain the one-time setup characteristic of the circuit, we define two sets of sub-circuits: $\mathbb{C}_t$ and $\mathbb{C}_{st}$. These sets include all feasible target-only checks $\sigma_{\theta,s}(r)$ and source-target differential checks $\sigma'_{\theta,s}(r, r')$ relevant to the particular use case. Here, $\theta$ is also variables fed into the circuit to ensure utmost flexibility, while $s$ acts as a selection bit, indicating the applicability of this edit-bound check. Consequently, the final edit-bound check circuit, denoted **EditCheck**$(r, r', \theta, s)$, operating on an old credential $r$ and its new counterpart $r'$, is formulated as: **Assert**$\big( \sum_{\sigma_{\theta,s} \in \mathbb{C}_t} (1 - \sigma_{\theta,s}(r)) \cdot s +$ $\sum_{\sigma'_{\theta,s} \in \mathbb{C}_{st}} (1 - \sigma'_{\theta,s}(r, r')) \cdot s == 0 \big)$. Note that both the parameters $\theta$ and the selection bits $s$ should be accessible to the verifier, ensuring the edit-bound check adheres to the correct protocol.

## 5.7 Achieving Strictly Linear Circuit Size

While the **Merge** technique delineated in Section 5.3.3 produces constraints linear in relation to the input size, consecutively merging LPS in a serial manner could culminate in a complexity of $O(N^2)$ constraints for documents with a maximum length of $N$ bytes. To enhance circuit efficiency, we introduce the **BatchMerge** strategy. This approach guarantees a strictly linear $O(N)$ constraint complexity for documents that span up to $N$ bytes in length. The main idea behind **BatchMerge** is to first gather all the LPSs that need to be merged in sequence. After collecting them, we then merge them all together using an enhanced version of the merge check process, detailed in Algorithm 4.

The transition from a 2-LPS merge check to a more comprehensive $n$-LPS merge check is similar to what we discussed in Section 5.3.3. We combine the target index with the byte value for each byte in the input LPS. Afterwards, we ensure the two batch multiplications are equal using randomness generated by an in-circuit random oracle. With minor alterations to the serialization procedure presented in Section 5.4, the entire IDE circuit can attain a linear circuit complexity in relation to the maximal document size. This optimization proves pivotal, especially when contemplating the scaling of JSON documents to vast dimensions, such as cataloging a scholar's complete publications within a singular document.

## 5.8 Put Everything Together

Using the modules we've described earlier, we now present a comprehensive circuit for Integrity-driven Editing (IDE) of JSON documents. The circuit begins with legitimacy and range checks for all inputted strings. Next, both the old and new credentials are serialized and encrypted. The circuit then compares these two versions to ensure they adhere to specified editing integrity rules. It's important to emphasize that the IDE circuit is designed to intrinsically encrypt both the old and new credentials using the same key to ensure that editors cannot modify the encryption key.

An essential step in finalizing the circuit is to validate that the in-circuit ciphertext matches the one computed externally. This ensures both serialization and encryption processes are accurate. Given the presence of dummy variables in our ciphertext variable array, we introduce a circuit, **AssertArray**$(a_{N_a}, b_{N_b})$, to validate the equality of two arrays, even if their lengths differ. It achieves this by padding the shorter array with zeros until both arrays reach a common length, $N = \max(N_a, N_b)$. The equality of the arrays is then verified using: **Assert**$\left( \sum_{i=0}^{N-1} \textbf{IsEqual}(a[i], b[i]) == N \right)$. This wraps up our discussion on the IDE circuit design. Each component has been meticulously refined for optimal performance, ensuring advanced capabilities with a reasonable constraint count. The complete IDE circuit can be found in Algorithm 5.

## 6 USE CASE

IDEA-DAC transforms credential management by emphasizing integrity-driven verification. It offers a more efficient approach than the traditional revoke-and-reissue method, guaranteeing that credential alterations uphold system integrity. DACs have diverse

applications, from employment to government services and medical records, underlining their practical significance.

According to the W3C's production rule [29] and verifiable credentials data model [31], credentials are serialized into JSON format. EXAMPLE 33 [32] displays a verifiable credential in a DID document linked to a resident card. Organizations can validate this by examining its JSON Web Signature and the issuer's public key. Technically aligned with the W3C standard, IDEA-DAC explores JSON serialization through a provable circuit, supporting multi-party edits. Specific entities, like past employers, can add details about job titles, employment spans, and roles. Professional organizations can input data on memberships, certifications, or accolades. Educational bodies can list degrees, grades, and academic milestones, while peers or clients might contribute testimonials or skill endorsements. Though individuals can't modify these entries in their DAC, they can control their disclosure, like showcasing recent testimonials while withholding older ones. This offers a balance: while individuals control their data's visibility, their reputation also relies on externally verified, immutable inputs.

IDEA-DAC, as elaborated in Appendix C, illustrates a multi-party editable credential system for DID. This protocol delineates the creation, reading, and updating of a credential document, akin to the *ethr-did* method on Ethereum [10]. While authorized parties can modify designated fields in the document, the permissions vary amongst different entities within these parties. In a university scenario, imagine the following Ph.D. profile acting both as a verifier of student status within educational institutions and as evidence of the educational level for job contexts:

```
{"program_status":"Ongoing","program_years":5,
"publications":[{"title":"XXX","year":2022}],
"student_id":"UNI421",
"duration":{"start":"08/01/18","end":"05/31/23"}}
```

The `program_status` field uses a specified set of strings to denote the student's current stage, and the `program_years` reflects the research duration. New articles can be added to the append-only `publications` without altering previous entries. Meanwhile, `student_id` provides a uniquely formatted identifier, and `duration` denotes the program's duration in a timeframe format.

## 7 EXPERIMENTS

We implemented the IDEA-DAC circuit for the PhD Profile use case described in Section 6 as our benchmark, and conducted experiments to assess our circuit's performance. Our algorithmic design is encapsulated within a comprehensive circuit, developed using the Gnark [9] framework. All experiments were conducted on a standard AWS EC2 `r5a.8xlarge` instance, equipped with 32 vCPUs and 256GB of memory. To evaluate the scalability of our circuit, we varied the document size by incrementing the maximum number of publications and then measured three key metrics across these sizes: the number of rank-1 constraints, the proving time, and verification time. Beyond the Encoding and Encryption checks, we incorporates five distinct integrity rules:

(1) **One of the Set:** Suitable for various data types, this rule limits modifications to a well-defined set of values. For example, within the `program_status` field, the permissible values are limited to "Approved", "Ongoing", "Graduated", and "Withdrawn".

(2) **Number Within Range:** Tailored for numeric entries, this rule ensures that a number remains within a specified interval. For example, the `program_year` should lie between 3 and 8.

(3) **Append Only:** This rule, apt for list or dictionary, facilitates the addition of new entries while disallowing the removal of existing ones. In a scholar's publication history context, while new publications can be appended, previous ones are immutable.

(4) **Meet Certain Format:** Targeted at strings, this rule mandates edits to align with certain format. For example, the student ID may necessitate an "AAA111" pattern for a specified institution.

(5) **Time Sensitive:** Time can be encoded by Unix timestamps. This rule ensures, for example, that a project's start date precedes its end date, and the end date precedes the current date.

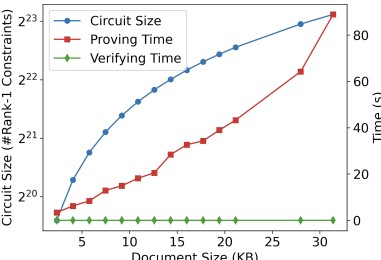

**Figure 3: Experimental results of IDEA-DAC, showcasing key metrics across varying credential document sizes.**

As depicted in Figure 3, the results highlight the algorithm's performance across document sizes ranging from 1KB to 33KB. We observe a clear linear correlation between the document size and the number of rank-1 constraints. Specifically, a document size of 1KB requires around $3 \times 10^5$ constraints, and there is an increase of approximately $5 \times 10^5$ constraints for every 2KB increment in the document size. Conversely, the proving time, which encompasses both constraint solving and proof generation, does not scale linearly. This non-linear behavior can be attributed to the use of ZK-friendly random oracles, such as MiMC [2], that are not amenable to parallel constraint solving. However, the time required for proof generation remains reasonable; for instance, a 32KB document takes just about 80 seconds. We underscore that our evaluations utilized the Groth16 [17] protocol over BN254 curve on a 32-vCPU machine—a basic configuration. Employing servers with enhanced CPU capabilities and leveraging advanced ZKP protocols will further optimize proving times, especially for large-scale applications. For example, a R1CS with $2^{22}$ constraints will only require 15s to prove using Orion [34] with a single CPU core. In terms of verification, Groth16 performs well, with the time ranging between 1ms and 2ms.

## 8 CONCLUSION

In the field of decentralized anonymous credentials (DACs) and Decentralized Identifiers (DIDs), updating credentials efficiently and with integrity is challenging. We present IDEA-DAC, a method for managing JSON credentials differently. IDEA-DAC utilizes Integrity-Driven Editing (IDE) with ZK-SNARKs and R1CS circuits for the editing process. Our R1CS design for JSON serialization optimizes efficiency. Tests indicate a linear connection between credential file size and constraint count. Overall, IDEA-DAC advances editing integrity in DACs, addressing large JSON dataset complexities. Future work can further build on these insights.

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

# A  CIRCUIT PSEUDOCODES

Additionally define the following two R1CS primitives for the naive Merge algorithm:

- $c \leftarrow$ **Lookup2**$(s_0, s_1, r_0, r_1, r_2, r_3)$: 2-bit lookup table. Output $r_0, r_1, r_2, r_3$ if $(s_0, s_1) = (0, 0), (1, 0), (0, 1), (1, 1)$, separately. This functionality can be achieved via a low-degree extension of the binary function, i.e., $f(s_0, s_1) = (1-s_0)(1-s_1)r_0 + s_0(1-s_1)r_1 + (1-s_0)s_1r_2 + s_0s_1r_3$. Simplifying the polynomial gives us a solution of using only 3 constraints: 1) $(r_3 - r_2 - r_1 + r_0) \cdot s_1 = t_1 - r_1 + r_0$; 2) $t_1 \cdot s_0 = t_2$; 3) $(r_2 - r_0) \cdot s_1 = c - t_2 - r_0$, where $t_1$ and $t_2$ are newly-added wires.

- $c \leftarrow$ **M**$(a_{[N]}, b)$: A multiplexer that selects the $b$-th element in $a_{[N]}$ where $b$ is also an in-circuit variable. We show a linear multiplexer implementation in Algorithm 7.

# B  FORMAL DEFINITION OF ZERO KNOWLEDGE PROOF (ZKP)

Formally, a ZKP consists of three algorithms $(P, V, S)$, where $P$ is the prover, $V$ is the verifier, and $S$ is the simulator. A ZKP has the following properties:

---

**Algorithm 1** Encoding a Number in ASCII String Format

---

1: **function** ENCODENUMBER(x, maxDigit)
    ▷ Input the decomposed decimals computed off-circuit
2:    $\text{len}\|\text{dec}_{[\text{maxDigit}]} \leftarrow \text{GetDecimal}(x, \text{maxDigit})$
3:    **RangeCheckDecimal**$\left(\text{dec}_{[\text{maxDigit}]}\right)$    ▷ Check 1
    ▷ Check the correctness of decomposition
4:    $\text{sum} \leftarrow 0, \text{remLen} \leftarrow \text{len}$
5:    $\text{isEnd}_{[maxDigit]} \leftarrow \textbf{NewArray}(\text{maxDigit})$
6:    **for** i = 0; i < maxDigit; i++ **do**
7:        $\text{isEnd}[i] \leftarrow \textbf{IsZero}(\text{remLen})$
8:        $\text{sum} \leftarrow \textbf{Select}(\text{isEnd}, \text{sum}, \textbf{Mul}(\text{sum}, 10))$
9:        **Assert**(**And**(isEnd, **IsNotZero**(dec[i])) == 0) ▷ Check 2
10:      $\text{sum} \leftarrow \textbf{Add}(\text{sum}, \text{dec}[i])$    ▷ No effect on dummy 0s
11:      $\text{remLen} \leftarrow \textbf{Select}(\text{isEnd}, 0, \textbf{Sub}(\text{remLen}, 1))$
12:   **Assert**(sum == x)    ▷ Check 3
    ▷ Get the ASCII-encoded LPS Representation
13:   $\text{res}_{[\text{maxDigit}]} \leftarrow \textbf{NewArray}(\text{maxDigit})$
14:   **for** i = 0; i < maxDigit; i++ **do**
15:      $\text{res}[i] \leftarrow \textbf{Select}(\text{isEnd}[i], D, \textbf{Add}(\text{dec}[i], 48))$
16:   **return** $\text{len}\|\text{res}_{[\text{maxDigit}]}$

---

**Algorithm 2** LPS Legitimacy Check

---

1: **function** LEGITIMACYCHECK($\text{aLen}\|a_{[N]}$)
2:   $\text{isEnd} \leftarrow 0, \text{numZero} \leftarrow 0, \text{numValid} \leftarrow 0$
3:   $c_{[N]} \leftarrow \textbf{NewArray}(N)$
4:   **for** i = 0; i < N; i++ **do**
5:      $d \leftarrow \textbf{IsDummy}(a[i])$
6:      $\text{isEnd} \leftarrow \textbf{Or}(\text{isEnd}, d)$
7:      $c[i] \leftarrow \textbf{Select}(d, \textbf{Sub}(i+1, \text{aLen}), \textbf{Sub}(\text{aLen}, i+1))$
8:      $\text{numZero} \leftarrow \textbf{Add}(\text{numZero}, \textbf{IsZero}(c[i]))$
9:      **if** i > 0 **then**
10:        $v1 \leftarrow \textbf{IsEqual}(c[i], \textbf{Add}(c[i-1], 1))$
11:       $v2 \leftarrow \textbf{IsEqual}(c[i], \textbf{Sub}(c[i-1], 1))$
12:       $\text{valid} \leftarrow \textbf{Select}(\text{isEnd}, v1, v2)$
13:       $\text{numValid} \leftarrow \textbf{Add}(\text{numValid}, \text{valid})$
14:   **Assert**(numValid == $N - 1$)
15:   **Assert**(**IsEqual**(numZero, 1)+**IsZero**(aLen) == 1)

---

**Algorithm 3** Hint-based Linear LPS Merge

---

1: **function** MERGE($\text{aLen}\|a_{[N_a]}, \text{bLen}\|b_{[N_b]}$)
    ▷ Input the merged string computed off-circuit
2:   $\text{cLen}\|c_{[N_a+N_b]} \leftarrow \text{Merge}(\text{aLen}\|a_{[N_a]}, \text{bLen}\|b_{[N_b]})$
3:   **Assert**(cLen == **Add**(aLen, bLen))    ▷ Check 1
4:   **RangeCheck**$\left(c_{[N_a+N_b]}\right)$    ▷ Check 2
    ▷ Check3: the correctness of LPS merge
5:   $r \leftarrow \textbf{RO}(\text{aLen}\|a_{[N_a]}, \text{bLen}\|b_{[N_b]}, \text{cLen}\|c_{[N_a+N_b]})$
    ▷ **IND** stands for **IsNotDummy**
6:   $\text{aMul} \leftarrow \prod_{i=0}^{N_a-1}[r - (256(i+1) + a[i])\textbf{IND}(a[i])]$
7:   $\text{bMul} \leftarrow \prod_{i=0}^{N_b-1}[r - (256(i+1+aLen) + b[i])\textbf{IND}(b[i])]$
8:   $\text{cMul} \leftarrow \prod_{i=0}^{N_a+N_b-1}[r - (256(i+1) + c[i])\textbf{IND}(c[i])]$
9:   **Assert**(**Mul**(aMul, bMul) == cMul)
10:  **return** $\text{cLen}\|c_{[N_a+N_b]}$

---

**Algorithm 4** Hint-based Linear Batch LPS Merge

---

1: **function** BATCHMERGE($l^{(0)}\|s^{(0)}_{[N_0]}, \ldots, l^{(n-1)}\|s^{(n-1)}_{[N_{n-1}]}$)
    ▷ Input the merged string computed off-circuit
2:   $\text{MaxLen} \leftarrow \sum_{i=0}^{n-1} N_i$
3:   $\text{oLen}\|o_{[\text{MaxLen}]} \leftarrow \text{Merge}(l^{(0)}\|s^{(0)}_{N_0}, \ldots, l^{(n-1)}\|s^{(n-1)}_{N_{n-1}})$
4:   **Assert**$\left(\text{oLen} == \textbf{Add}(l^{(0)}, l^{(1)}, \ldots, l^{(n-1)})\right)$    ▷ Check 1
5:   **RangeCheck**$\left(o_{[\text{MaxLen}]}\right)$    ▷ Check 2
    ▷ Check3: the correctness of LPS merge
6:   $r \leftarrow \textbf{RO}(l^{(0)}\|s^{(0)}_{[N_0]}, \ldots, l^{(n-1)}\|s^{(n-1)}_{[N_{n-1}]}, \text{oLen}\|o_{[\text{MaxLen}]})$
    ▷ **IND** stands for **IsNotDummy**
7:   $\text{cumLen}_i \leftarrow \sum_{j=0}^{i-1} l^{(j)}$
8:   $\text{sMul}_i \leftarrow \prod_{j=0}^{N_i-1}[r - (256(j+1+\text{cumLen}_i) + s[j])\textbf{IND}(s[j])]$
9:   $\text{oMul} \leftarrow \prod_{i=0}^{\text{MaxLen}-1}[r - (256(i+1) + o[i])\textbf{IND}(o[i])]$
10:  **Assert**(**Mul**(sMul_0, sMul_1, \ldots, sMul_{n-1}) == oMul)
11:  **return** $\text{oLen}\|o_{[\text{MaxLen}]}$

---

**Algorithm 5** Integrity-driven Editing Circuit

---

1: **function** IDE($r_{old}, r_{new} : \mathcal{D}, k : \mathbb{F}, c_{old}, c_{new} : [\mathbb{F}], \theta, s : [\mathbb{F}]$)
2:   $c'_{old} \leftarrow \textbf{Encrypt}(k, \textbf{Compress}(\textbf{Encode}(r_{old})))$
3:   $c'_{new} \leftarrow \textbf{Encrypt}(k, \textbf{Compress}(\textbf{Encode}(r_{new})))$
    ▷ Assert the equivalence of two arrays with unequal length
4:   **AssertArray**($c'_{old} == c_{old}$)
5:   **AssertArray**($c'_{new} == c_{new}$)
6:   **EditCheck**($r_{old}, r_{new}, \theta, s$)

---

**Algorithm 6** Naive LPS Merge

---

1: **function** NAIVEMERGE($\text{aLen}\|a_{[N_a]}, \text{bLen}\|b_{[N_b]}$)
2:   $l_1 \leftarrow \text{aLen}, l_2 \leftarrow \text{bLen}, \text{outLen} \leftarrow \textbf{Add}(l_1, l_2)$
3:   $\text{out}_{[N_a+N_b]} \leftarrow \textbf{NewArray}(N_a + N_b)$
4:   $a_{[N_a+N_b]} \leftarrow \text{pad}(a_{[N_a]}, \mathbf{0}_{[N_b]})$
5:   $b_{[N_a+N_b]} \leftarrow \text{pad}(\mathbf{0}_{[N_a]}, b_{[N_b]})$    ▷ Avoid index overflow
    ▷ Extract the output array
6:   **for** i = 0; i < $N_a + N_b$; i++ **do**
7:      $\alpha \leftarrow \textbf{IsZero}(l_1), \beta \leftarrow \textbf{IsZero}(l_2)$
8:      $\text{out}[i] \leftarrow \textbf{Lookup2}(\alpha, \beta, a[i], \mathbf{b[i+N_a\text{-aLen}]}, a[i], 0)$
        ▷ **b[i+$N_a$-aLen]** computed by a linear multiplexer
9:      $l_1 \leftarrow \textbf{Select}(\alpha, 0, \textbf{Sub}(l_1, 1))$
10:     $l_2 \leftarrow \textbf{Lookup2}(\alpha, \beta, l_2, \textbf{Sub}(l_2, 1), l_2, 0)$
11:  **return** $\text{outLen}\|\text{out}_{[N_a+N_b]}$

---

**Algorithm 7** Linear Multiplexer

---

1: **function** M($x_{[N]}, \text{idx} : |\mathbb{F}|$)
2:   $\text{logN} \leftarrow \text{logCeil}(N)$    ▷ $\lceil \log_2 N \rceil$
3:   $\text{res}_{[2^{\text{logN}}]} \leftarrow \text{pad}(x_{[N]}, \mathbf{0}_{[2^{\text{logN}}-N]})$
4:   $\text{idxBin}_{[\text{logN}]} \leftarrow \textbf{ToBinary}(\text{idx}, \text{logN})$
5:   **for** i = 0; i < logN; i++ **do**
6:      **for** j = 0; j < $2^{(\text{logN}-i-1)}$; j++ **do**
7:        $\text{res}[j] \leftarrow \textbf{Select}(\text{idxBin}[i], \text{res}[2j+1], \text{res}[2j])$
8:   **return** res[0]

- **Completeness:** If the statement is true, the honest verifier (that is, one following the protocol properly) will be convinced of this fact by an honest prover. Formally, for any $x, w$ such that $x \in L_w$ and any verifier strategy $V^*$,

$$\Pr[(P(x, w), V^*(x)) = 1] = 1.$$

- **Soundness:** If the statement is false, no cheating prover can convince the honest verifier that it is true, except with some small probability. Formally, for any $x \notin L_w$, any prover strategy $P^*$, and any $w'$,

$$\Pr[(P^*(x, w'), V(x)) = 1] \leq negl(\lambda),$$

where $negl(\lambda)$ is a negligible function.

- **Zero-knowledge:** If the statement is true, no verifier learns anything other than this fact. This is formalized by showing that every verifier has some simulator that, given only the statement to be proved (and no access to the prover), can produce a transcript that "looks like" an interaction between the honest prover and the verifier in question. Formally, for any $x, w$ such that $x \in L_w$, any verifier strategy $V^*$, and any $w'$,

$$\{(P(x, w), V^*(x))\} \approx \{(S(x, w'), V^*(x))\},$$

where the approximation symbol $\approx$ denotes computational indistinguishability.

## C A TENTATIVE SYSTEM DESIGN OF IDEA-DAC

In this section, we introduce a potential design of the whole IDEA-DAC system.

### C.1 Identity Configuration

In the IDEA-DAC system, the Identity Configuration process is crucial for transparent user identity establishment, involving both users and editors. Protocol "SetupIdentity" (Protocol 8) outlines essential steps for identity creation. It takes inputs $r$, a public number, and the user's key pair $P_k, S_k$, producing a new key pair, $P_{ID}$ and $S_{ID}$.

---

**Protocol 8** SetupIdentity

---

**Input:** $r, P_k, S_k$
**Output:** $P_{ID}, S_{ID}$
**if** *choose to expose account* **then**
    $P_{ID} = P_k$
    $S_{ID} = S_k$
**else**
    $S_{ID} = S_k \oplus r$
    $P_{ID} = PubGen(S_{ID})$
**return** $P_{ID}, S_{ID}$

---

Users can either reveal their account or stay anonymous. If revealed, the key pair remains as $(P_{ID} = P_k, S_{ID} = S_k)$. For anonymity, $S_{ID}$ is formed by XORing $S_k$ with $r$ and $P_{ID}$ is then obtained using the PubGen function on $S_{ID}$. After this, the algorithm provides the identity key pair, $P_{ID}$ and $S_{ID}$, for all in the IDEA-DAC system, ensuring secure, private exchanges.

### C.2 Editor Configuration

In the IDEA-DAC system, editor configuration is achieved through two protocols: SetupEditorDAO (Protocol 10) and SetupEditor (Protocol 9). The former sets the foundation for the Editor DAO through steps like proposing, voting, and establishing the Editor DAO.

---

**Protocol 9** SetupEditor

---

**On behalf of editor $E_i$:**
$\sigma_i, Mkp, Mkr = MPC(\phi_i)$ # $\phi_i$ is the qualification of $E_i$, and $\sigma_i$ is the edit limitation
Upload Mkr or $P_e$ to smart contract if needed
**Algorithm MPC:**
Input: Each user's qualification $\phi_i$ as secret inputs
Output: Edit Limit El, Merkle Path Mkp, Merkle Root Mkr
Give each editor an appropriate edit limitation
Build Merkle trees for different edit limitations
Generate new common public key $P_e$

---

In the IDEA-DAC system, the Manager DAO starts the editor configuration process. A member, denoted as $i$, proposes a new credential type $R = (Des, m, w)$, with $Des$ being the content description, $m$ the MPC code, and $w$ the edit circuit. The proposal undergoes a voting process among Manager DAO members. If it gains approval, the members reach a consensus on the Edit Circuit Setup, co-sign the Edit Verifier contract (VC), and upload it. The Manager DAO then establishes the Editor DAO, with interested editors joining the computation $m$ and getting their limitations and related information. The updated Merkle Root for the credential type and editing limitation is then uploaded.

Protocol 9, termed SetupEditor, focuses on an individual editor $E_i$. It determines the editor's edit limitation, $\sigma_i$, as well as their Merkle Path (Mkp) and Merkle Root (Mkr) using the MPC algorithm, based on their qualification, $\phi_i$. The MPC then assigns edit limitations, constructs Merkle trees for these, and creates a new public key, $P_e$. Editors may upload the Mkr or $P_e$ to a smart contract as needed.

Together, these protocols set up the Editor DAO in the IDEA-DAC system, ensuring a decentralized approach to editor management.

### C.3 User Configuration

In the IDEA-DAC system, user setup is handled by both a protocol, SetupUser, and a circuit, SetupUserCircuit.

The SetupUser protocol (Protocol 11) focuses on user $U_i = (Pk_i, Sk_i)$. Its primary goal is to generate and manage unique keys for each of the user's credentials. When a user has a credential $r_i$, they create a distinct encryption key, $K_{r_i}$, with the KeyGen function. This key is then signed using their secret identity key, $S_{ID}$, resulting in signature, $\tau$. The composed pair $K_{r_i}||\tau$ is forwarded to the Editor DAO. In response, the Editor DAO provides the user with a threshold signature, $\tau'$, for the encryption key, $K_{r_i}$. Subsequently, the user crafts a SetupUserCircuit proof, which is uploaded to the blockchain.

The SetupUserCircuit (Circuit 12) is a zero-knowledge proof circuit, ensuring the validity of the user's key and signature without exposing any private information. It uses public inputs like the committed encryption key, $K_{c_{ri}}$, and the EditorDAO public key, $P_{DAO_{ri}}$. Secret inputs encompass the encryption key, $K_{ri}$,

---

**Protocol 10** SetupEditorDAO

---

**Propose:**

Member $i$ in Manager DAO raises proposal for a new kind of credential $R = (Des, m, w)$:

$Des$ := Content description

$m$ := MPC code

$c$ := Edit circuit

**Voting:**

Members in Manager DAO will vote to decide whether to accept

**Uploading:**

Proposer run $Setup(1^\lambda, c)$

Majority of Members agree on a version of Edit Circuit Setup

Majority of Members threshold sign on the Edit Verifier contract (VC) and upload

**Form Editor DAO:**

Each inclined editor joins the computation $m$ and gets her limitation and other information (refer to 9)

Manager DAO then upload updated Merkle Root in the corresponding credential type and editing limitation

---

and the EditorDAO signature on it, $\tau$. The circuit's task is to confirm a statement. This statement checks the signature's authenticity and the commitment of the encryption key. It is only validated if both the signature and commitment evaluations succeed $(True == SigVerify(K_{ri}, \tau) \bigwedge K_{c_{ri}} == Commit(K_{ri}))$.

In essence, this approach ensures a secure and private way for users to set up their credentials in the IDEA-DAC system.

---

**Protocol 11** SetupUser

---

**On behalf of user** $U_i = (Pk_i, Sk_i)$

**for** each credential $r_i$ **do**

    $K_{r_i} = KeyGen()$

    $\tau = Sign_{S_{ID}}(K_{r_i})$

    Send $K_{r_i} || \tau$ to editors' DAO

    Receive

$\tau' = ThresholdSign_{Editors}(K_{r_i})$

    Generate SetupUserCircuit Proof and upload on chain

**On behalf of Editors**

Receive $(K_{r_i} || \tau, P_{ID})$

Verify $\tau$

**if** Success **then**

    Store mapping from $P_{ID}$ to $K_{r_i}$

**else**

    Abort

---

**Circuit 12** SetupUserCircuit

---

**Input:** Public: Committed encryption key $K_{c_{ri}}$, EditorDAO Public Key $P_{DAO_{ri}}$

Secret: encryption key $K_{ri}$, Signature of EditorDAO on encryption key $\tau$

**Statement:**

$True == SigVerify(K_{c_{ri}}, \tau) \bigwedge$

$K_{c_{ri}} == Commit(K_{ri})$

---

## C.4 Editing Process

The Editing Process in the IDEA-DAC system ensures editors can alter users' encrypted credentials, safeguarding privacy and upholding data integrity, as articulated in Protocol 13 (Edit) and illustrated via two circuits: EditCircuit(General) (Circuit 14) and EditCircuit(FastTrack) (Circuit 15).

When an editor, recognized by the public identity $P_{ID_e}$, intends to edit user $P_{ID}$'s encrypted credential $r_i$, they firstly decrypt it utilizing the disclosure key $K_{ri}$ to access the original content $C$. Post-editing to form modified content $C'$, and depending on blockchain Merkle Root changes since the last edit, either an EditCircuit(General) proof or an EditCircuit(FastTrack) proof is generated and later verified in the Profile Contract.

EditCircuit(General) (Circuit 14) entails both public and secret inputs. Public ones encompass the prior credential $r$, the updated $r'$, user $P_{ID_u}$, edit limit $\sigma$, Merkle Root $root$, and key commitment $K_{c_{ri}}$. Secret inputs include the editor's secret identity $S_{ID_e}$, original content $C$, revised content $C'$, Merkle Path $path$, and the encryption key $K_{ri}$. This circuit authenticates a statement, validating the editor's identity, Merkle Root, edit limit, key commitment, and the content's encryption and decryption processes.

This structured, rigorous approach warrants precise, confidential editing of encrypted credentials within the IDEA-DAC system while maintaining data veracity and user privacy.

---

**Protocol 13** Edit

---

Assume Editor $P_{ID_e}$ wants to edit the user $P_{ID}$'s encrypted credential $r_i$. Closure Key for this user's credential is $K_{ri}$.

Original content $C = Dec_{K_{ri}}(r_i)$

Modify the content and get modified content $C'$

**if** Merkle Root on chain not changed since last editing **then**

    Generate EditCircuit(General) Proof and get verified in Profile Contract

**else**

    Generate EditCircuit(FastTrack) Proof and get verified in Profile Contract

---

**Circuit 14** EditCircuit(General)

---

**Input:** Public: Old Credential $r$, New Credential $r'$, User $P_{ID_u}$, EditLimit $\sigma$, Merkle Root $root$, KeyCommit $K_{c_{ri}}$

Secret: Editor $S_{ID_e}$, Old Content $C$, New Content $C'$, Merkle Path $path$, encryption key $K_{ri}$

**Statement:**

$P_{ID_e} = PubGen(S_{ID_e}) \bigwedge$

$P_{ID_e}$ and $path$ can generate $root$ $\bigwedge$

$C - C' \le \sigma \bigwedge$

$K_{c_{ri}} == Commit(K_{ri}) \bigwedge$

$r == Enc_{K_{ri}}(C) \bigwedge$

$r' == Enc_{K_{ri}}(C')$

---

The EditCircuit(FastTrack) (Circuit 15) offers a more efficient alternative to EditCircuit(General), optimized for reduced computational demands. By bypassing the verification of the editor's identity and Merkle Root, it necessitates a more concise set of inputs. Within

---

**Protocol 17** ProfileContract

$MR$ denotes a dictionary
$\{key : EditLimit\ \sigma,$
$value : [Merkle\ Root\ r , PublicKey\ P_e]\}$
$R$ denotes a dictionary
$\{key : (UserID\ P_{ID_u}, CredentialID\ R_{id}),$
$value : Encrypted\ Credential\ Content\ C\}$
$Addresses$ denotes a dictionary
$\{key : address, value : EditLimit\ \sigma\}$
$Keys$ denote a dictionary
$\{key : (UserId, CredentialId),$
$value : commit(ClosureKey\ K)\}$

**Function SetupUser:**
**Input:** SetupUserProof $p$, SetupUserWitness $w$, CredentialID $R_{id}$
Check $w.P_e == MR[w.\sigma].P_e$
Check $True == Verify(p, w)$
if all pass, $Keys[(P_{ID_u}, R_{id})] = w.K_{c_{ri}}$

**Function Edit:**
**if** sender not in $Addresses$ **then**
 **Input:** EditCircuitProof $p$, EditCircuitPublicWitness $w$, CredentialID $R_{id}$
  Check $MR[w.\sigma] == w.root$
  Check $R[(w.P_{ID_u}, R_{id})] == w.r$
  Check $w.K_{c_{ri}} == Keys[(w.P_{ID_u}, R_{id})]$
  Check $Verify(p, w) == True$
  If any of the checks failed, **Reject**
  If all pass, $R[(w.P_{ID_u}, R_{id})] = w.r'$
**else**
 **Input:** EditCircuitProof(FastTrack) $p$, EditCircuitPublicWitness(Fast track) $w$, CredentialID $R_{id}$
  Check $R[(w.P_{ID_u}, R_{id})] == w.r$
  Check $w.K_{c_{ri}} == Keys[(w.P_{ID_u}, R_{id})]$
  Check $Verify(p, w) == True$
  If any of the checks failed, **Reject**
  If all pass, $R[(w.P_{ID_u}, R_{id})] = w.r'$

---

**Circuit 15** EditCircuit(FastTrack)

**Input:** Public: Old Credential $r$, New Credential $r'$, User $P_{ID_u}$, EditLimit $\sigma$, KeyCommit $K_{c_{ri}}$
Secret: Old Content $C$, New Content $C'$, encryption key $K_{ri}$
**Statement:**
$C - C' \leq \sigma \wedge$
$K_{c_{ri}} == Commit(K_{ri}) \wedge$
$r == Enc_{K_{ri}}(C) \wedge$
$r' == Enc_{K_{ri}}(C')$

---

EditCircuit(FastTrack), the validation is confined to checking the edit limit, key commitment, and the accurate encryption and decryption of the content.

To wrap up, the Editing Process within the IDEA-DAC framework, encompassing the Edit protocol and both EditCircuit forms, delivers a robust mechanism for safely and adeptly editing encrypted user credentials. It upholds user privacy and the integrity of the data. The FastTrack circuit variant offers a computational advantage, especially beneficial when edits are made without alterations to the Merkle Root since the previous edit.

---

**Circuit 16** ValidationCircuit

**Input:** Public: Credential $r$, EditorDAO Common Pubkey $P_e$, Criteria $\epsilon$, Committed encryption key $K_{c_{ri}}$
Secret: Content $C$, encryption key $K_{ri}$
**Statement:**
$K_{c_{ri}} == Commit(K_{ri}) \wedge$
$r == Enc_{K_{ri}}(C) \wedge$
$C\ satisfy\ \epsilon$

---

## C.5 Profile Contract and Verification

The IDEA-DAC system's Profile Contract (Protocol 17) manages user credentials, edit rules, and cryptographic keys, with directories for Merkle Roots, encrypted data, and editor identities. It primarily executes the SetupUser and Edit functions for user credential handling.

The system's unique feature is the ValidationCircuit (Circuit 16), allowing users to verify compliance with criteria without revealing raw credentials. This circuit uses public inputs like the credential $r$ and EditorDAO public key $P_e$, and secret ones like content $C$. It checks the validity of encrypted content and adherence to criteria, ensuring users can confirm their alignment to standards without exposing plaintext credentials.

