# OpenReview forum: "IDEA-DAC: Integrity-Driven Editing for Accountable Decentralized Anonymous Credentials"
_ACM.org/TheWebConf/2024/Conference — TheWebConf24_

### Official Review · Reviewer_XXbe · 2023-11-21

**Novelty:** 5
**Technical Quality:** 5

**Review:**

This paper present IDEA-DAC, a new method for enabling Integrity-Driven Editing in decentralized anonymous credentials (DACs), in order to address shortcomings of the current revoke-and-reissue paradigm. The proposed method, leveraging R1CS circuits' properties, enables constraint-adhering credential amendments through edit-bound conformity checks, while maintaining DAC security and privacy properties. Importantly, IDEA-DAC requires a one-time setup and deployment for homogeneous JSON credentials.

While I'm not extremely familiar with the area, I find that the paper was relatively easy to follow and well motivated; the authors seem to properly relate their work to prior systems, while clearly highlighting the gap which they aim to fill. Also, despite the sheer number of (necessary) definitions and notations, I think most of them are presented in a rather intuitive way, easing readability.

My first question concerns the maximum length of variables, which allows for IDEA-DAC's one-time setup. I assume that picking an extremely large threshold to accommodate lengthy edits might affect the system's efficiency; is this true? If so, how is this maximum length optimally selected and which one did you use for your use-case evaluation? Also, while I understand the authors' assumption on JSON documents' static nature, i.e., maintaining data types, I would like to ask how would the system behave or what would it require to operate if, for instance, the JSON credential changes format, data types etc.

Moreover, I would expect the evaluation to be more thorough, given that IDEA-DAC can facilitate different modifications. Specifically, I understand that the edits made to the example, use-case credential are to add more publications, meanwhile increasing the document size. Additional edits could be adding new dictionary fields, changing string/numeric values etc. Are these incorporated in the current evaluation and, if not, would they affect the presented performance results?

- Pros
    + Enables precise, integrity-driven editing in DACs
    + Requires one-time setup for homogeneous JSON documents

- Cons
    + Evaluation could possibly be more comprehensive

**Questions:**

- How is the maximum length predetermined and how does it affect the system's efficiency?

- What are IDEA-DAC's requirements for accommodating new credential formats, e.g., changing types?

- What precise edits were carried out in the evaluation?

**Reviewer Confidence:**

1: The reviewer's evaluation is an educated guess

**Scope:**

3: The work is somewhat relevant to the Web and to the track, and is of narrow interest to a sub-community

---

### Official Review · Reviewer_324e · 2023-11-22

**Novelty:** 5
**Technical Quality:** 5

**Review:**

In this paper, the authors introduce IDEA-DAC, a novel methodology designed to facilitate Integrity-Driven Editing (IDE) for Accountable Decentralized Anonymous Credentials. Overall, the paper is well-crafted and presents concepts in a clear manner.

The abstract goes right away to the point and might not be easy to follow. It has many technical details and might be a roadblock for many potential readers. To improve this, you might want to avoid acronyms and try to explain your proposal assuming that readers have no prior knowledge on the field. Recall that WebConf is not a crypto conference so the writing style should be adjusted accordingly.

A key concern revolves around the practical feasibility of the proposed methodology in real-world environments. As highlighted in Section 7, the proving time is non-linear and time-consuming. Considering that users typically lack resources such as 32 vCPUs and 256GB of memory, the paper should address the question of whether this schema is currently feasible in practical scenarios.

Additionally, given that the proposal introduces a new DAC schema for enabling users to edit JSON documents, the absence of source code is a notable limitation. Including the source code would not only allow readers to validate the proposed results and deploy the solution in a real environment but also encourage contributions to future research in the field.

Minor things:

-Enlarge Table 1 to utilize available empty spaces, enhancing readability.

-Consider merging Sections 6 and 7 as their separation seems unnecessary.
-
Review and avoid the use of abbreviations for improved clarity.

**Questions:**

-Using [24] as a baseline, there is a notable lack of detailed information regarding the practicality of the solution in terms of machine capabilities and the performance of the proposed schema. It would be beneficial to clarify the minimum specifications required for all parties involved in deploying the proposed IDEA-DAC.

-The experiments were conducted on a "standard AWS EC2 r5a.8xlarge instance, equipped with 32 vCPUs and 256GB of memory." However, this configuration significantly deviates from a typical computer setup. In comparison to [24], where experiments were carried out on a "2021 Intel i9-11900KB CPU with 8 physical cores and 64GiB RAM," the need for such powerful hardware should be justified.

-Given that the paper introduces a new DAC schema, the inclusion of the source code is crucial. As the proposal itself is a fundamental aspect, providing access to the source code would enhance the transparency and reproducibility of your work.

**Reviewer Confidence:**

3: The reviewer is confident but not certain that the evaluation is correct

**Scope:**

3: The work is somewhat relevant to the Web and to the track, and is of narrow interest to a sub-community

---

### Official Review · Reviewer_uChu · 2023-11-23

**Novelty:** 4
**Technical Quality:** 3

**Review:**

A method of IDEA-DAC, integrated driver editing (IDE) with zk-snark and R1CS circuits for editing process is proposed. The efficiency of R1CS design for JSON serialization is optimized, and the editing integrity of dac is improved.

**Questions:**

1, the experimental design is insufficient, the lack of horizontal comparison data of similar software;
2. It is mentioned that the proof time can be further optimized by using a more powerful CPU and advanced ZKP protocol. Does the current scheme encounter performance bottlenecks when dealing with large-scale JSON certificates?
3, please explain how to support selective disclosure and predicate proof of credentials, and how to ensure the anonymity and privacy of users;
4. When describing completeness, the paper mentions the identified limitations in the existing system. Please describe the limitations in detail.

**Ethics Review Description:**

NULL

**Reviewer Confidence:**

4: The reviewer is certain that the evaluation is correct and very familiar with the relevant literature

**Scope:**

4: The work is relevant to the Web and to the track, and is of broad interest to the community

---

### Official Review · Reviewer_bc4R · 2023-11-23

**Novelty:** 6
**Technical Quality:** 6

**Review:**

**Summary**

This paper presents IDEA-DAC, a decentralized anonymous credential system that is designed to support the modification of JSON credential documents of variable length, while ensuring the integrity of these edits. The proposed system employs R1CS and ZK-SNARKs to implement edit-bound conformity checks, ensuring adherence to predefined rules and compliance with integrity standards. One important aspect of this work is that the proposed circuits can support JSON documents up to a specific size without the need for recompilation, setup, or smart contract redeployment.

**Comments**

- The paper is well written in general, easy to follow and understand. I enjoyed reading this paper.
- It's an interesting and important work in the area of DAC systems. It proposes a solution to the issue of credential editing while ensuring their integrity, without the need for the system to revoke and reissue the credentials. I find that this work contributes to advancements in the area of credential management.
- One issue with this paper is that some of the complementary materials included in the appendix are, indeed, very useful for better understanding and following the paper. I would definitely suggest having some algorithms and protocols in the main body of the paper, but I understand that this may be difficult due to the space limitation.
- The experimental evaluation presents key metrics such as the circuit size, and proving and verifying time, for credential documents of different sizes. In that regard, I would like to see a comparison of this system’s performance with other DAC systems (for example those in Table 1), and also an exploration of the trade-offs between any additional verification overheads that incur in this system in comparison to the costs involved in existing systems that revoke and reissue credentials.

**Questions:**

It seems that the proving time does not scale linearly. Which operations are responsible for this increase, and can these be further optimised? It seems to me that the proving time might become unacceptably high If the JSON credentials increase significantly in size.

**Reviewer Confidence:**

2: The reviewer is willing to defend the evaluation, but it is likely that the reviewer did not understand parts of the paper

**Scope:**

4: The work is relevant to the Web and to the track, and is of broad interest to the community

---

### Official Review · Reviewer_H1SQ · 2023-12-01

**Novelty:** 5
**Technical Quality:** 5

**Review:**

Decentralized Anonymous Credential (DAC) systems enable users to verify specific attributes of their identity without revealing the complete identity to the verifier. This paper focusses on an interesting problem. Traditionally DAC systems allow updates to existing credentials in a revoke-and-reissue approach which increases the verification overhead. This paper presents IDEA-DAC that enables edits directly to a JSON credential document utilizing zero knowledge proofs built on rank-1-constraint-system.

I enjoyed reading this paper, and find the research problem interesting and novel. The work is important in the space of privacy-centric authentication because IDEA-DAC helps with reducing computational redundancy and verification overhead. It also ensures integrity in the editing process of the credentials.

Pros
------
* The paper works on an important and timely problem.
* The paper clearly describes the IDEA-DAC mechanism and provides a use case to explain the end-to-end functionality.
* The verification time performance is encouraging.

Cons
-------
* The paper does not compare its performances with a state-of-the-art system or a traditional DAC system.
* The paper does not discuss other formats of verifiable credentials.

Comments
------------
This paper discusses an important problem, and proposes a system that may have significant impact on the DAC ecosystem. The paper does an excellent work in explaining the IDEA-DAC system. I have few questions for the authors -
* In Section 7 there is no baseline to compare IDEA-DAC’s performance against. Could you show how IDEA-DAC’s performance improves over traditional revoke-reissue credential systems? How IDEA-DAC is better than closely related work like Candid, Coconut etc?
* While I understand that IDEA-DAC focusses on JSON formatted credential documents, the discussion on other types of credential formats is missing. What types of other credential formats are there? How IDEA-DAC would need to be extended to support those formats?
* What are the limitations of IDEA-DAC?

**Questions:**

Please see review comments.

**Reviewer Confidence:**

2: The reviewer is willing to defend the evaluation, but it is likely that the reviewer did not understand parts of the paper

**Scope:**

3: The work is somewhat relevant to the Web and to the track, and is of narrow interest to a sub-community

---

### Decision · Program_Chairs · 2024-01-22

**Decision:**

Accept

**Comment:**

# Summary

 Decentralized Anonymous Credential (DAC) systems enable users to verify specific attributes of their identity without revealing their complete identity to the verifier. This paper specifically focuses on the problem of how to update existing credentials. Prior DAC systems allow updates to existing credentials in a revoke-and-reissue approach which increases the verification overhead. This paper presents IDEA-DAC that enables edits directly to a JSON credential document utilizing zero knowledge proofs built on rank-1-constraint-system. This is useful in smart contract applications (and thus relevant to TheWebConf).

 # Strengths

 + The paper addresses an important problem.
 + Verification time experiments show significant improvement.
 + Paper is well-written and easy-to-follow, even for non-experts.

 # Weaknesses

 - Evaluation could be improved to traditional DAC techniques.

 # Recommendation

 Overall, the reviewers felt that this paper addresses an important problem and proposes an interesting solution that pushes the state-of-the-art forward. In addition, the reviewers also appreciated that the authors engaged in the discussion process, as that answered several of the reviewers questions/concerns on the paper. Therefore, given the strengths and the fit for TheWebConf, I recommend accepting this paper.

 ---